# Whole-Blood Longitudinal Molecular Profiling Maps the Road of Graft Versus Host Disease (GVHD)

**DOI:** 10.3390/cancers17050802

**Published:** 2025-02-26

**Authors:** Merav Bar, Mohammed El Anbari, Darawan Rinchai, Mohammed Toufiq, Dhanya Kizhakayil, Harshitha S. Manjunath, Rebecca Mathew, Irene Cavattoni, Sabine Forer, Marco Recla, Hani Bibawi, Ahmad Alater, Reem Yahia, Clarisa Brown, Nancy L. Miles, Phuong Vo, Davide Bedognetti, Sara Tomei, Ayman Saleh, Chiara Cugno, Damien Chaussabel, Sara Deola

**Affiliations:** 1Fred Hutchinson Cancer Research Center, Clinical Research Division, Seattle, WA 98109-1024, USA; 2Sidra Medicine, Research Department, Doha P.O. Box 26999, Qatar; 3General Hospital of Bolzano, Hematology and BMT, 39100 Bolzano, Italy; 4Sidra Medicine, Pathology Department, Doha P.O. Box 26999, Qatar; 5Sidra Medicine, Hematology-Oncology Division, Doha P.O. Box 26999, Qatar

**Keywords:** GVHD, biomarkers, transcriptional fingerprint assay (TFA)

## Abstract

With a method that measures the abundance of 264 genes from a few drops of fingerstick-collected blood, we analyzed blood changes in patients after allogeneic hematopoietic cell transplantation for the first time at high frequency: every week/second week. By correlating the results with patients’ health status, we discovered important biological processes conducive to graft versus host disease (GVHD). Such genes may suggest a way to prevent GVHD.

## 1. Introduction

Graft versus host disease (GVHD) is a common complication after allogeneic hematopoietic cell transplantation (allo-HCT) with high rate of short- and long-term morbidly and mortality [1,2,3,4,5]. Despite extensive research, few predictors of GVHD have been incorporated into clinical practice to date [6,7,8,9,10]. A major limitation in finding relevant GVHD biomarkers is the complexity of allo-HCT, implying layers of multi-scale system interactions that promote deviation from the normal immunobiological milieu and responses. Moreover, markers of GVHD are difficult to measure using conventional techniques, which typically consider only a few variables at limited timepoints. A number of attempts have been made to measure transcriptomic perturbation during GVHD. While interesting transcriptome longitudinal data have been generated using animals [11], most of the data captured from humans were only cross-sectional and therefore not accurate representations of the complex immune milieu after allo-HCT [12,13]. Peripheral blood gene expression is usually performed on 3–15 mL of blood volume, after venipuncture. High-frequency sampling is therefore unfeasible in such fragile patients as it would require an overall considerable amount of blood, additional to what already used for routine clinical follow-up.

Thus, there is a need for novel approaches that capture longitudinally complex dynamic data and provide flexible and low-cost solutions to this problem.

We implemented here a transcriptional fingerprint assay (TFA) using multiplex microfluidics q-PCR capturing a wide range of systemic immune perturbations [14]. This assay requires a minimal volume of blood, obtainable by a micro-invasive procedure such as blood fingerstick.

The set of 264 genes comprising the panel investigated in this study were selected based on a fixed repertoire of blood transcriptional modules (BloodGen3) [15]. The construction of this repertoire itself was informed by clustering patterns observed among a set of 985 unique blood transcriptome profiles, spanning 16 different diseases or physiological states (including a wide range of autoimmune/inflammatory diseases, infectious diseases, solid organ transplantation, cancer, and pregnancy). Notably, the BloodGen3 repertoire has been extensively characterized functionally, and bioinformatics resources have been developed to aid with the interpretation and visualization of the data analyzed using this modular framework. As such, it is uniquely tailored to the evaluation of complex immune perturbations through frequent profiling of gene expression signatures [14,16,17]. This approach has been successfully applied to stratify the prognosis of patients with autoimmune and infectious diseases [18,19,20,21,22].

Using this approach, we were able to study, for the first time, allo-transplanted patients longitudinally at high frequency (weekly or be-weekly) for a period of up to 430 days post-allo-HCT. Novel and divergent mechanisms associated with the onset and severity of GVHD were identified.

The results obtained may provide important insights on unknown mechanisms of GVHD genesis, steroid refractoriness, and responsiveness to immunomodulatory therapies, and can be used to modulate therapeutic intervention accordingly.

## 2. Materials and Methods

### 2.1. Study Population

Adult patients undergoing related or unrelated bone marrow or peripheral blood allo-HCT and related donors were recruited to this study. Enrollment was open to every patient eligible for allo-HCT (“all allo-HCT comers”), and the only exclusion criterion was a previous allo-HCT. The protocol was approved by the IRB of the participating clinical centers and all participants gave written informed consent.

Here, we report the results of 31 consecutive patients enrolled in Fred Hutchinson’s Hospital in Seattle, WA, USA, and the General Hospital of Bolzano, Italy, from August 2018 through September 2019 (this study is ongoing). Adult sibling donors consented to low-volume blood sample collection at at least one timepoint prior to donation (optional collections every 6 months for 2 years).

### 2.2. GVHD Classification and Grading

Acute GVHD (aGVHD) was classified according to the Glucksberg modified criteria [23] and chronic GVHD (cGVHD) was classified and graded according to the 2014 NIH Consensus criteria [24].

### 2.3. Blood Sampling

Micro-quantities (50 μL) of whole blood were collected once prior to transplant conditioning, then weekly from engraftment time until day 100 post-transplantation and every 2 weeks thereafter until 2 years post-transplantation. The 1-week and 2-week sampling intervals were chosen empirically to reconcile a high-frequency sampling program with a patient- and resource-feasible schedule. Samples were collected either in the hospital by a nurse or by the patients or their caregivers using a fingerstick procedure (https://www.youtube.com/watch?v=xnrXidwg83I, accessed on 18 February 2025) [25]. Briefly, blood (1 volume) was collected through capillarity in a microtube prefilled with Tempus RNA preservative solution (2 volumes) [17]. After blood collection, the microtube was shaken vigorously and immediately stored at −20 °C at home (in regular home freezers) or at −80 °C in the hospital.

Samples from patients who were not local to the transplant center area were shipped in batches stored at 4 °C from the patient’s home to the referring center, and periodically shipped on dry ice to Sidra Medicine, Doha, Qatar, for analysis. The robustness of sampling, storing, and shipping methods was verified in preliminary experiments [26].

### 2.4. Clinical Data Annotation

For each sample timepoint, the study nurse or coordinator recorded coded annotations of clinical, laboratory, and therapy details, including medication lists and patient diaries, as well as GVHD and infection assessments. The data were collected and stored in a dedicated REDCap database and checked manually for quality against the source electronic medical records by the clinical investigators.

### 2.5. RNA Extraction and Sample Quality Assessment

Tempus Spin RNA Isolation kits (ThermoFisher, Waltham, MA, USA) were used to isolate and purify RNA from blood collected in the capillary tubes using a protocol adapted for small-volume blood collection [27]. RNA quality and quantity were evaluated using a NanoDrop OneC Microvolume UV-Vis spectrophotometer (ThermoFisher, Waltham, MA, USA) and, when appropriate, verified using a Bioanalyzer (Agilent Technologies, Santa Clara, CA, USA). Samples were regarded as generally acceptable for analysis according to the following criteria: (A260:A280) purity > 1.7 and RNA integrity number > 6. RNA QC was performed for every sample, and only high-quality samples were included in the Fluidigm analysis, either meeting the above-mentioned purity and integrity criteria or showing discrete RNA bands after Bioanalyzer verification.

### 2.6. Fluidigm High-Throughput PCR Data Generation and Processing

Expression of 264 immune-related genes [14] was measured in each sample using the Fluidigm BioMark high-throughput q-PCR system. To verify the absence of contamination, qPCR reactions included non-template and blank negative controls, both of which showed no amplification. The Fluidigm Real Time PCR software (version 4.5.2 Standard BioTools, former ‘Fluidigm’, San Francisco, CA, USA) was used to merge the technical runs, and threshold cycle (Ct) values were processed using the Partek Genomic Suite (version 7.18; Partek, Saint Louis, MO, USA). Genes with Ct values outside the detectable range were recorded as missing values. Each gene was then normalized to a validated house-keeping gene pool (*DOCK2*, *EEF1A1*, *FAM105B* (*OTULIN*), *FTL*, *MYL6*, *MYL12B*, *RPS10*, and *RPS25*) transformed to −∆Ct (negative delta Ct) and anti-log2, and batch-corrected with ComBat [28]. All raw and curated data, including metadata, are in Appendix A.

### 2.7. Modular Analysis

The 264 TFA gene panel was selected as a representation of a broader panel called BloodGene3 [15]. The BloodGene3 repertoire was constructed on the basis of patterns of co-clustering observed for a reference collection of 16 pathological and physiological states, encompassing the blood transcriptome profiles of 985 individual subjects [14]. Information about the composition of the modules and functional annotations can be accessed via https://drinchai.shinyapps.io/BloodGen3Module/, accessed on 18 February 2025. Briefly, the architecture of this repertoire is pyramidal, based on data-driven co-clustering of genes grouped for their similar activity profiles. The basis of the pyramid is composed of 1528 genes, grouped 4 by 4 in 382 “modules” with common activity profiles and a common functional annotation (example “neutrophil”, or “cytokines”). A further clustering at the upper level of the pyramid groups the modules in “aggregates”, again based on similar module activity profiles.

In our data analyses, we used the TFA panel at the “gene”, “module”, and “aggregate” levels. A complete list of genes included in the TFA along with their module membership and grouping in “aggregates” is provided in Appendix A.

### 2.8. Statistical Analyses

The following analyses refer to 31 consecutive patients enrolled in the study and assigned retrospectively to the “GVHD” or “non-GVHD” cohort, according to the manifestation of any sign of acute or chronic GVHD during the post-transplant course.

Pre-transplant GVHD prediction and group comparison gene expression analyses were performed using *t*-tests and hierarchical clustering (HC) in R [29]. *p*-values < 0.05 and gene expression fold changes >1.5 were used to identify differentially expressed genes (DEGs). Significant genes identified in the pre-transplantation GVHD prediction analysis were tested longitudinally with a linear mixed model in R using the lme4 package [30] measuring significance in time and group (GVHD/non-GVHD) interactions.

The longitudinal analyses of the GVHD and non-GVHD cohorts post-transplantation were also performed in R using the change data point (CDP) detection [31] and the penalized generalized estimating equation (PGEE) methods [32], which were chosen to highlight significant differences when analyzing limited numbers of subjects and missing values. Additionally, GVHD data were categorized into three outcomes for analysis: (i) pre-GVHD, (ii) active-GVHD (defined as acute/chronic GVHD that was clinically measurable), and (iii) post-non-active GVHD.

CDP analysis evaluates every series independently, highlighting significant changes in gene expression for each post-transplantation course individually. In this analysis, “change points” are measured as abrupt and statistically significant changes in the variance and mean gene expression. This method is also used in the modeling and prediction of time series in different fields [33]. Due to its “series” specificity, we chose the CDP method to reduce the bias of the small non-GVHD control cohort. This method also highlights only robust gene fluctuations, increasing the specificity and consistency of transcriptome expression data.

The PGEE method was used as it is suited to the analysis of high-dimensional data with multiple covariates. This method allows evaluation of correlations in discrete outcome datasets with missing data values, simultaneously estimating coefficients and performing feature selection. The optimal tuning parameter was chosen by cross-validation.

## 3. Results

### 3.1. Patients Analyzed

Thirty-one patients with a pre-transplantation sample and reaching at least 169 days of clinical follow-up post-allo-HCT (169–672 days; median 513) were included in the pre-transplantation gene prediction analysis. Patient characteristics are summarized in Table 1. Samples collected longitudinally post-allo-HCT were available for eighteen patients, but three were excluded; two due to a clinically uncertain GVHD diagnosis and one because of poor sample quality. Longitudinal blood samples from the remaining 15 patients were annotated with clinical features including GVHD status, disease status, and infections. Samples with unclear GVHD annotations were removed from the analysis. In summary, a total of 31 samples pre-allo-HCT and 245 post-allo-HCT were suitable for analyses, collected until a median of 215 days after allo-HCT (70–430 days) (Table 2, Appendix A for clinical annotations and data).

### 3.2. RNA Quality Assessment

A general assessment of RNA quality was performed on the first 215 samples obtained. Of those, 170 (84%) were self-collected by patients and 45 (16%) by a research nurse during hospital stays or clinic visits. Of these samples, 193 (90%) passed the QC and were suitable for analysis by Fluidigm multiplex q-PCR. Of the twenty-two samples that failed the QC, eight were from a single patient who generally performed poorly in collections (these samples were excluded from the analyses), three were collected into microtubes without Tempus solution, and eleven represented accidental failures that were randomly distributed among patients.

The average quantity of capillary self-collected RNA was 767.6 ± 704.8 ng and the average purity (A260:A280) was 2.2 ± 0.9.

### 3.3. Pre-Transplantation Gene Expression May Predict the Risk of GVHD

Overall, 26 patients developed GVHD (12 aGVHD only, 3 late aGVHD, 1 overlap syndrome, 5 acute and chronic GVHD, and 5 cGVHD only) and 5 patients did not.

Using the “lme4” linear mixed model, 31 available pre-transplantation samples were tested for association with subsequent GVHD development using a *t*-test and a longitudinal analysis of the following parameters: (1) time; (2) group (GVHD/non-GVHD), and (3) interactions between (1) and (2).

Among patients who developed GVHD, three genes (*GPSM3*, *LRG1,* and *EPHX4*) were significantly downregulated in the pre-transplantation sample and two genes (*IFI35* and PLK3) showed borderline (<1.5-fold) upregulation. In the GVHD group, several other genes (111) were found to be upregulated (>1.5 fold), although the change in gene expression did not reach the level of statistical significance (Figure 1(AI)). Of note, 12 of these genes were highly upregulated (>4-fold), and their transcriptome was related to B cells, T lymphocytes, monocytes, IFN, TNF, other cytokines, and neutrophil activation (Figure 1(AII)). IFN genes were consistently represented by multiple IFN modules, emerging as promising candidates for distinguishing GVHD development post-transplantation.

Longitudinal analyses confirmed that *EPHX4* and GPSM3 distinguished the cohorts for all the parameters (time, group, and interaction) measured during the post-transplantation period. *EPHX4* expression increased over time in both the GVHD and non-GVHD groups, although the increase was significantly greater in the non-GVHD group (fixed effect estimate: 2.05). *GPSM3* expression decreased over time, although the decrease was significantly greater in the non-GVHD group (fixed effect estimate: −0.95). The changes in *LRG3* expression were significant only for the parameter “time”, and again, the decrease was significantly greater in the non-GVHD group (Figure 1B).

Longitudinal analysis of IFN modules and individual genes revealed very distinct patterns in the GVHD and non-GVHD cohorts. Furthermore, when tested individually, the changes in expression of several IFN-related genes proved statistically significant over ”time” (*NT5C3*, *CCR1* and *LBA1*) for “group” (*NT5C3*) and “time–group interactions” (*IFI27*, *CCRI* and *LBA1*) (Figure 1C).

### 3.4. Longitudinal Expression Signatures of GVHD Activity

All 245 samples, collected from 15 patients with (n = 11) or without (n = 4) a clinical diagnosis of GVHD, were used in the post-alloHCT analysis of gene expression. Samples were divided into “non-GVHD” (n = 48), collected from 4 patients never diagnosed with GVHD, and “GVHD” (n = 197), collected from 11 patients who were diagnosed with acute and chronic GVHD during their post-transplantation course. Furthermore, to incorporate the variable of “time” in the analysis, and to focus on GVHD biological activity, “GVHD” samples were divided into three groups: “pre-GVHD”, collected prior to clinical diagnosis of GVHD (n = 29); “active-GVHD”, collected during the time of clinically active GVHD (measurable using the GVHD grading scale) (n = 55); and “post-GVHD” collected after clinical manifestations of GVHD were resolved (n = 113). All samples were classified according to the number of days before, during, or after GVHD onset and GVHD descriptions were added (Appendix A: “post-tx phenotypes” worksheet and Figure 2A).

First, all “active-GVHD” samples (n = 55) were compared with samples collected from patients without GVHD (“non-GVHD”; n = 48) and patients with “pre-GVHD” (n = 29). The “non-GVHD” and “pre-GVHD” samples were grouped in this analysis to form a more precise biological control for the status of GVHD activity.

In total, 41 DEGs were identified by comparison of the cohorts (*p* < 0.05). By clustering these genes according to the GVHD type (acute, chronic mild, chronic moderate, late acute/overlap syndrome), we detected a skewed pattern of signatures. “TNF”, “inflammation”, “IFN”, “cytokines/chemokines”, and “erythroid” genes were overexpressed in GVHD samples, while a varied signature of overexpression of very diverse modules was detected in non-GVHD samples (Figure 2B).

To further explore these signatures, we clustered all “active-GVHD” samples according to module aggregates. This analysis revealed very defined “acute” and “chronic mild” GVHD clusters, while “chronic moderate”, “late acute”, and “overlap” GVHD clustered together (Figure 2C). The “acute GVHD” cluster was defined by an IFN aggregate (A28), inflammation aggregates (A33,35), other aggregates with no known definition (25, 30, 31 and 32), and modules pertaining to the following functions: TNF (in A18), neutrophils/neutrophil activation (in A38), and prostanoids (in A34). Cytotoxicity modules (in A2) defined more “chronic mild GVHD”, while protein synthesis and B cells (in A1) and plasma cells (in A27) were upregulated in the “chronic moderate, late acute and overlap GVHD” cluster. The “erythroid” aggregates (A36 and 37) appeared again in the “chronic mild” and “acute GVHD” clusters.

These analyses highlighted clear signatures of DEGs associated with GVHD activity in different GVHD subtypes.

### 3.5. CDP Analysis Reveals Consistent Patterns of Abrupt Variations and “IFN” and “Erythrocyte” Modules Associated with GVHD Onset

To analyze transcript fluctuation patterns over time, we applied the CDP analysis to the complete allo-HCT samples series, including the pre-transplant timepoints. Specifically, the CDP analysis reveals abrupt variations in data series and is applicable only to series of ≥10 sequential data points. Datasets meeting this criterion were available for 12 patients (2 non-GVHD, 6 aGVHD, and 4 with both acute and cGVHD), while 3 patients were excluded (2 non-GVHD, 1 with both acute and cGVHD) (Table 2).

Of the 264 genes analyzed, abrupt changes in the dataset were detected for 25 genes belonging to 11 (out of 66) modules (Figure 3B,C). The “protein synthesis” module M10.2 (*HBB*, *RPS12,* and *OAZ1*) *CTSS* gene (cathepsin S, associated with MHC class II antigen presentation linked to the “inflammation” module) and *TXNIP* gene (thioredoxin interacting protein, regulator of cellular redox signaling linked to “neutrophils”) appeared consistently in all patients, including non-GVHD patients (Figure 3B, first two plots), and some pre-transplantation samples. This observation indicates a transcriptome pattern with a very robust tendency for abrupt variations, but not necessarily linked to the allo-HCT course.

Other significant CDP modules pertaining to “cell cycle”, “neutrophils/neutrophil activation”, “erythroid” lineages, “interferon”, and “chemokines/cytokines”, and single genes for “inflammation”, “B cells”, and “monocytes” appeared only in the “acute” and “acute and chronic” GVHD groups, but not the “non-GVHD” group (summarized in Figure 3C).

To distinguish transcripts associated with GVHD onset, we shortlisted all genes appearing exclusively at the timepoint preceding GVHD or at GVHD onset (the first GVHD timepoint). IFN-related genes (*LY6E*, and *IFI16*) and SLC2A14 (cytokines/chemokines) emerged in aGVHD, and erythroid-related genes (*SIAH2*, *SERF2*) in both aGVHD and cGVHD (Figure 3B, second and third rows of plots). Of note, erythroid-related genes tended to maintain sustained expression during GVHD.

Other genes and modules, including the protein synthesis transcript M10.2, emerged at various timepoints during active GVHD, but also in non-active GVHD.

This analysis revealed the identity of genes displaying consistently abrupt changes after allo-HCT and highlighted abrupt changes in some genes pertaining to IFN and erythroid lineage before and during the onset of GVHD.

### 3.6. PGEE Analysis of GVHD and Non-GVHD Cohorts Confirms Signatures Related to GVHD

To elucidate the gene fluctuations associated with GVHD, not only looking at abrupt changes, but rather at overall trajectories of transcripts over time, we performed a PGEE analysis of the GVHD and non-GVHD cohorts. We chose this type of statistical analysis since it is suited for series of samples displaying multiple covariates, while is able to correct for missing values. Of note, while CDP analysis considers every series (patient) independently, PGEE analysis is applied to groups. All 15 patients with longitudinal samples were included in this analysis; however, to decrease the number of missing data values and increase the robustness of the method, we consolidated the datasets by setting a threshold at timepoint 19, covering a median of 197 days post-transplantation (timepoints 20–37 were excluded, representing only two patients with samples collected for >1 y post-transplantation; see Figure 2A or Figure 3A for visual representation). This analysis revealed the following genes that significantly distinguished GVHD over time: *ABHD5* (unknown function, module TBD), *CTSS* (inflammation), *TXNIP* and *DEFA4* (neutrophil/neutrophil activation), the IFN-related genes *IFI16* and *IFI44*, and *OAZ1* and *RPS12* (M10.2 protein synthesis). Surprisingly, genes that displayed abrupt variations in non-GVHD samples in the CDP analysis (*CTSS*, *TXNIP*, *OAZ1,* and *RPS12*) also clearly distinguished the GVHD clinical course, either by a different form of fluctuation or by their abundance (Figure 4A left panels). In the same analysis performed according to modules, modules 10.1 (IFN) and 10.4 (neutrophil activation) significantly distinguished the GVHD clinical course (Figure 4B left panels).

Finally, we conducted PGEE analysis of the 41 DEGs identified in the previous *t*-test analysis of active GVHD versus non-GVHD samples (shown in Figure 2B). *ABHD5*, *CTSS*, *IFI44*, *TXNIP,* and the M10.2 protein synthesis module *OAZ1 RPS12*, (this time including *HBB*) were confirmed to significantly distinguish active GVHD, and *RBM38* (A37, “erythroid” aggregate) was newly identified in this analysis (Figure 4A,B right panels).

Thus, this analysis provides robust evidence that genes related to protein synthesis, inflammation, neutrophil/neutrophil activation, and interferon and erythroid pathways represent biomarkers of the course of GVHD after allo-HSCT and verified the signatures obtained in the previous statistical analyses.

### 3.7. Candidate Signatures of “GVHD Onset”

Gene fluctuations related to GVHD’s clinical course may reflect the natural history of GVHD; however, these patterns may be confounded by GVHD therapy-driven gene expression programs or perturbations related to infections and other post-transplantation-associated events. To identify a GVHD onset signature free of GVHD-therapy-related biases, we grouped all GVHD events with a pre- and post-GVHD onset timepoint (Appendix A, worksheet “post-tx phenotypes”, GVHD days pre-/post-onset) and aligned acute and cGVHD episodes over time. For aGVHD onset analysis, we used nine aGVHD episodes (seven primary and two flares) in eight patients. For cGVHD analysis, we used five episodes in four patients, all with previous aGVHD. One patient developed late aGVHD, which evolved to moderate cGVHD by day 119 post-transplantation. After this cGVHD had resolved, the patient was diagnosed with mild cGVHD at day 324 (the two episodes were counted separately). All recurrent aGVHD and cGVHD episodes had at least one (1–19 days prior, median 5) or two (20–35 days prior, median 22) prior GVHD-free timepoints.

We compared the following signatures (significant for GVHD in the previous analyses) in the pre- and post-GVHD onset timepoints: “IFN” (A28 aggregate), “cytotoxicity” (A2 aggregate), “neutrophil” (M15.35), and “neutrophil activation” modules (M10.4 in A38 aggregate) and the “erythroid” aggregate (A37 aggregate) (Figure 5). As a reference control, we also evaluated the same signatures in each patient pre-transplantation in the donor cohort and in the non-GVHD group (Figure 5).

Clear trends of differences between GVHD-related gene fluctuations and the non-GVHD baseline gene expression were confirmed in all modules, which was consistent with the results of the longitudinal analyses.

With respect to pre- and post-GVHD onset differences, the number of datapoints was not sufficient for a statistical trend analysis. However, the “IFN” aggregate and “neutrophil activation” module peaked before the onset of aGVHD, but not before cGVHD. The shape of this curve thus implicates these genes as candidates for aGVHD prediction. Furthermore, the “neutrophil” module clearly peaked after aGVHD onset, probably reflecting the effects of steroid therapy. These findings indicate that the biological pathway of neutrophil activation precedes GVHD and is independent of neutrophil abundance.

TLR2 innate immunity/HMGB1 DAMP receptor was also associated with early GVHD onset, with a higher peak in aGVHD and a lower peak in cGVHD (data available on request). In pre-cGVHD-onset samples, more modest changes in gene expression were detected, including those of the “cytotoxicity” and “erythroid” signatures.

### 3.8. Candidate Signatures of “Pure GVHD”

Finally, to separate transcriptional fluctuations related to GVHD from signatures triggered by infections or related to relapse occurrence, we excluded from the dataset all timepoints coinciding with active infections, until infection clearance (13 timepoints in 9 patients), and coinciding with relapse status (3 timepoints in 2 patients) (Appendix A, worksheet “post-tx phenotypes”). We repeated the PGEE analysis with this dataset and confirmed the gene signatures that previously emerged in the first PGEE analyses, except for *ABHD5*, *RBM38,* and *IFI16.* Additional genes emerged with significantly different expression in the GVHD versus non-GVHD cohorts: *CDC34*, *TRAK2*, *SERF2*, belonging to “erythroid” modules; *GPB1* and *STAT1* belonging to “IFN” modules; and the single genes *MSN* (neutrophils), *GPSM3* (inflammation), *TLR2* (cytokines/chemokines), and *PFAAP5* (protein synthesis). At modular level, neutrophil activation (M10.4), along with cell cycle (M12.15) and cytokines/chemokines (M13.16), confirmed a GVHD-distinctive pattern based on the abundance of gene transcripts over time (Appendix A).

In synthesis, these data reiterate the candidature of “IFN”, “erythroid”, “protein synthesis”, “inflammation”, “neutrophils/neutrophil activation”, and “cytokines/chemokines” transcriptional patterns for GVHD identification, either represented by single genes or by modular trajectories.

Notably, the “erythroid” signature was consistently found to distinguish GVHD from non-GVHD samples throughout all the analyses. Although an abundance of hemoglobin in random samples can be the result of uneven lysis of red blood cells, this artifact was excluded by analysis of the expression of the hemoglobin gene (*HBB*; belonging to module 10.2 (A9) labeled “protein synthesis”), which clustered consistently with the other genes (*OAZ1* and *RPS12*) in the same module and aggregate. Thus, we confirmed that the erythrocyte modules (A36 and 37) represent a specific GVHD signature, predominantly sub-defining features of acute and mild chronic GVHD (Figure 2).

## 4. Discussion

Transcriptome analyses in humans have been used to track mostly cross-sectional data for the course of GVHD post-allo-HCT [11,12,13], and limitations in sampling capacity in allo-HCT recipients have restricted the identification of reliable genomic markers of GVHD.

The aim of our study is to overcome such limitations by allowing frequent sampling and identification of strong associations between genomic findings and clinical events. The same TFA method has been used to study several infectious diseases [34,35,36] and autoimmune conditions [19,37] by utilizing a well-defined set of immune-related genes that represents a snapshot of the immune system [18]. The selection of the gene repertoire analyzed in this study was not tailored to GVHD biomarkers, but rather was based on co-clustering patterns measured across various immunological and physiological conditions [14]. We therefore analyzed the post-allo-HCT immune events using this pre-established “compass”, with the advantage of an unbiased analysis, and the limitation of only a selected choice of genes representing known GVHD-associated pathways. However, this limitation may be easily overcome in the future by customizing the multiplex q-PCR platform of the TFA and adding more specific GVHD-related markers.

To test the relevance of the TFA method for the identification of signatures of allo-HCT, we compared it with well-known GVHD pathways, most of which were identified in mouse models [38]. The initiating events in aGVHD consist of DAMP/PAMP-mediated tissue damage, neutrophil invasion, and innate and adaptive immune activation with IFN production. Our analysis of the available transcripts revealed an association of neutrophil activation (M10.4) and adaptive immune activation (IFN, A28) signatures with aGVHD onset, both exhibiting a peak before the onset. TLR2 innate immunity/DAMP receptor was also associated with early GVHD onset and a neutrophil abundance (presumably reflecting the effects of steroid second-line therapy) immediately after. These data mimic GVHD biology and implicate M10.4 and A28 as pre-GVHD candidate biomarkers.

We also discovered novel immune signatures distinguishing GVHD from non-GVHD and clustering different forms of GVHD. Clear patterns distinguished GVHD from non-GVHD samples with a relative abundance of selected signatures, including “IFN”, “TNF”, “chemokines/cytokines”, “inflammation”, and “erythroid cells”, as opposed to a variety of mixed transcripts characterizing the non-GVHD healthy status. Acute and chronic mild GVHD samples clustered separately, while overlap/late acute and chronic moderate samples clustered together.

The post-allo-HCT GVHD course was also distinguishable from a healthy non-GVHD status in longitudinal analyses through the shape or abundance of selected modules (M10.1 “IFN” and M10.4 “neutrophil activation”) and single genes related to “protein synthesis”, “IFN”, “neutrophil/neutrophil activation”, “erythroid cells”, and “inflammation”. By excluding infection-related episodes and relapse status, “IFN”-related single genes *IFI44* (M8.3 “type I interferon”), *GPB1, and STAT-1* (M10.1 “interferon”) were confirmed significant, while the whole “IFN” module 10.1 (composed by “*GPB1*”, “*IFI35*”, “*STAT-1*”, and “*ZBP1*” genes) was not confirmed, suggesting that *IFI35* and *ZBP1* genes rather reflect a microbial- or post-relapse-induced IFN expression.

“Erythroid”-, “inflammation”-, “neutrophil/neutrophil activation”-, and “protein synthesis”-related gene signatures, as well as the *TLR2* gene, also distinguished GVHD traits without infection/relapse interference, along with modules M10.4 “neutrophil activation”, M12.15 “cell cycle”, and 13.16 “protein synthesis”.

IFN gene signatures were present in abundance in patients developing GVHD, even before transplantation, but not in their donors, raising the possibility that a patient-specific IFN-axis germline composition may facilitate the occurrence of GVHD. Several IFN-related genetic variants influence microbial immune responses, autoimmune disease predisposition, and response to solid tumors [39,40,41] and may also occur in GVHD development if considering host APC IFN production [42].

In contrast, patients who did not develop GVHD showed higher levels of *GPSM3*, *LRG1,* and *EPHX4* pre-transplantation. *GPSM3* (G protein signaling modulator 3) is an MHC-regulated gene [43] bearing alleles that protect against severe autoimmune diseases. *LRG1* (leucine-rich α-2 glycoprotein 1) is a myelopoiesis modulator known to facilitate CD34 and myeloid progenitor growth by antagonizing the effects of transforming growth factor (*TGF-β*) [44], but has also been reported to be upregulated in GVHD patients [45]. In accordance with previous reports, *LRG1* expression increased in the GVHD cohort during the post-transplantation course. EPHX4 (epoxide hydrolase 4) is a protein with hydrolase activity, expressed in hematopoietic stem cells, T cells, and NK cells, and other undefined functions.

The main limitation of our study is the restricted number of subjects, particularly in the non-GVHD cohort, and the lack of a validation cohort. In addition, none of the subjects developed any extreme events (e.g., Grade-IV and steroid-resistant GVHD). Finally, the limited number of subjects did not have the statistical power to analyze subcategories of acute and chronic GVHD in depth. The study is currently ongoing with the target of 50 patients for each cohort (GVHD, non-GVHD, and donors), which is likely to overcome such limitations and hopefully enable prevention strategies on both acute and chronic organ-specific GVHD complications.

Despite the limited number of subjects, our study is built on a large dataset of 31 pre- and 245 post-alloHCT samples frequently collected over time, and representing, to our knowledge, the largest human sample collection in this field.

## 5. Conclusions

In conclusion, we demonstrated that TFA can be used to analyze the expression of immune-related genes in allo-HCT patients and to identify specific pathways associated with, and possibly predictive for, GVHD development. Among the significant genes, “IFN”, “erythroid”, “protein synthesis”, “inflammation”, “neutrophils/neutrophil activation”, and “cytokines/chemokines” transcriptional patterns, either represented by single genes or by modular trajectories, allowed the identification of GVHD traits, free from infection and relapse as confounding biases.

Upon completion of this study, we aim to provide a comprehensive definition of the transcriptome signatures related to acute and chronic GVHD and to validate the most important ones prospectively. Such signatures might be implemented clinically to predict GVHD before their occurrence and to modulate therapeutic interventions accordingly.

## Figures and Tables

**Figure 1 cancers-17-00802-f001:**
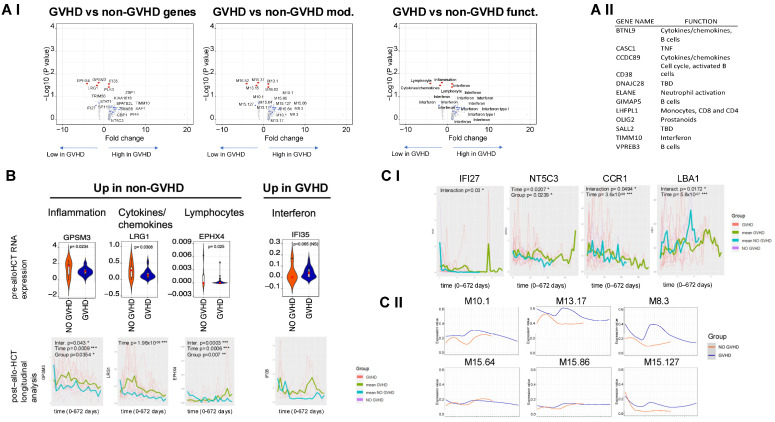
Differentially expressed genes pre-transplantation in the GVHD and non-GHVD cohorts. (**AI**) Volcano plots showing pre-transplantation expression of genes in the GVHD versus non-GVHD cohorts displayed for *p*-value (Y) and log change (X). Gene symbols and relative modules and functions are plotted from left to right. Genes with *p* < 0.05 and log change > 1.5 are shown in red and IFN-related genes with log change >1.5 are shown in blue. In grey: all other genes with log change > 1.5 and *p* > 0.05. (**AII**) Genes not reaching *p* < 0.05 but with log change >4. (**B**) Violin plots of the significant pre-transplantation genes (in (**A**)) are plotted (top) and displayed over time (bottom) after transplantation; data represent individual patient and average measurements. The significant parameters in the longitudinal analysis (GVHD/NO GVHD “group” “time” and “interaction”) are annotated for each gene at the top of the graph. (**C**) Expression of IFN-related genes and modules in the cohorts over time. (**I**) Statistically significant IFN-related genes after transplantation identified in the longitudinal analysis; (**II**) all IFN modules included in the analyses. Each module is composed of four genes (extensive explanations in Rinchai et al. [21] https://doi.org/10.1002/ctm2.244, accessed on 18 February 2025 and all modules are listed in Appendix A). * = *p* < 0.05; ** = *p* < 0.01; *** = *p* < 0.001.

**Figure 2 cancers-17-00802-f002:**
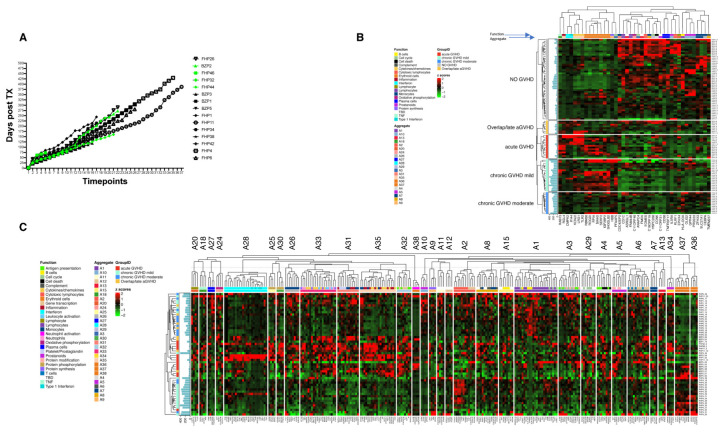
Differentially expressed genes post-transplantation in the GVHD and non-GHVD cohorts. (**A**) Correlation matrix of days post-transplantation versus progressive timepoint number for all samples included in the longitudinal analysis, graphically showing the consistency in the frequency of samples. (**B**) Hierarchical clustering of ACTIVE GVHD and NO GVHD (NEVER + PRE GVHD) differentially expressed genes, clustered according to the type of GVHD. The complete set of data with groups and clinical annotations is available in Appendix A. Column clustering (also in (**C**)): the first bar-row represents the modules color-coded for each function (legend on the left); the second bar-row represents the module aggregation (summarized in Appendix A). (**C**) Hierarchical clustering of ACTIVE GVHD samples clustered according to the module aggregates. For module aggregate definition, see Appendix A.

**Figure 3 cancers-17-00802-f003:**
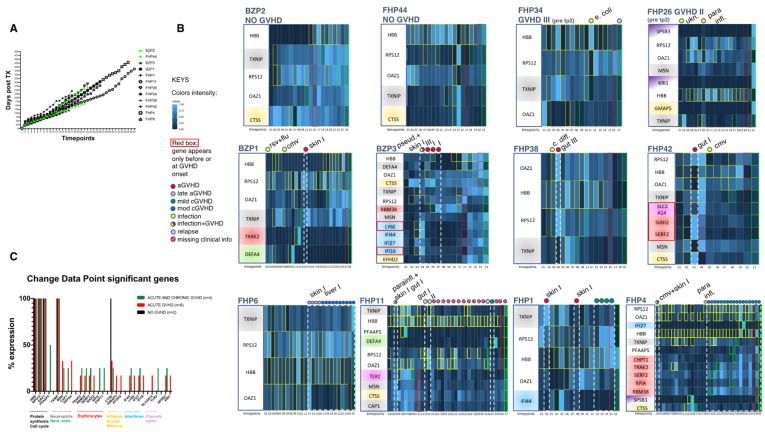
Change data point (CDP) analysis. (**A**) Correlation matrix of days post-transplantation versus progressive timepoint number for all samples included in the CDP analysis. (**B**) Longitudinal distribution of genes identified as significant in the CDP analysis of NO GVHD (first two plots, upper left) and GVHD patients. Patients without any active GVHD are shown in the second 2 plots in the upper left, while GVHD active episodes captured in the longitudinal analyses are contoured by white dashed boxes in the plots and described with GVHD organ-specific description and grading on top of the plots, along with infection episodes, as per the legend. Twenty samples of patient FHP11 with missing clinical info (symbol as per legend) are shown in the analysis, but not considered in the interpretation. Yellow boxes within the plots represent the timepoints of significant change (X axis) for each gene. Red boxes represent significant changes in gene clusters, while green boxes denote the end of the analysis. The intensity scale of gene expression is shown in the legend. Gene annotations (Y axes) are color-coded according to module membership as represented in (**C**). Genes significantly changing expression exclusively before (only the timepoint before) GVHD and during the first timepoint of GVHD onset are boxed in red on the Y axes. Genes showing significant changes in the NO GHVD plots were not considered in GVHD-related genes. (**C**) Graphical map of the distribution of CDP genes, and pertaining modules, measured collectively in the cohorts. The Y axis shows the percentage of expression of each gene/module in all samples (i.e., 100 = expressed in 100% of patient group samples).

**Figure 4 cancers-17-00802-f004:**
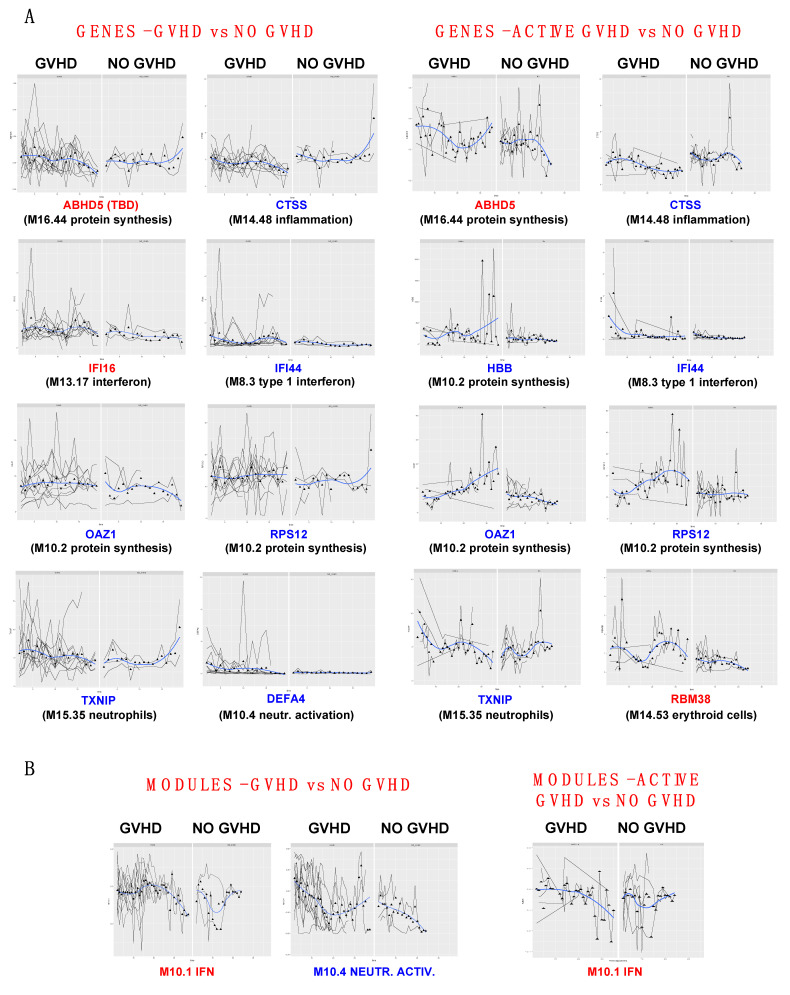
Penalized generalized estimating equation (PGEE) analysis. (**A**,**B**) PGEE significant genes and modules comparing GVHD versus NO GVHD cohorts or ACTIVE GVHD versus NO GVHD cohorts as indicated. For every gene and module, the analyzed cohorts are displayed in parallel to facilitate the visual distinction of the gene/module fluctuation. An additional PGEE analysis was run after the exclusion of infection- and relapse-related timepoints from the dataset, as described in the section “Candidate signatures of pure GVHD”. The genes and modules confirmed significant in this additional analysis are shown here in blue font, while the genes/modules not confirmed significant are annotated in red. Further genes and modules appearing significant in this additional analysis are shown in Appendix A. Black lines: gene expression of each subject over time; blue curves: smoothed trend curve of gene expression over time; triangles: mean expression at each time point.

**Figure 5 cancers-17-00802-f005:**
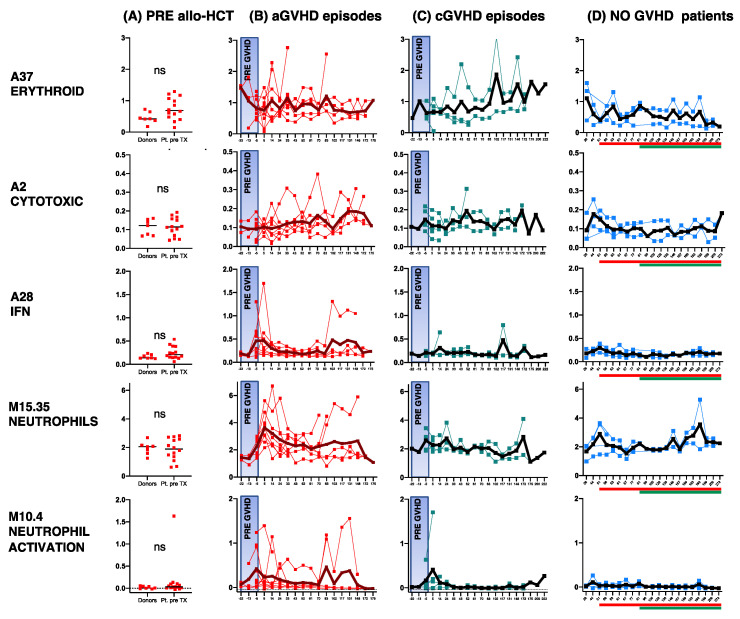
GVHD pre-onset candidate signatures. (**A**) Gene expressions of aggregates and modules, found significant in the previous analyses, are displayed (pre-allo-HCT (n = 7 donors and n = 14 patients)) along with their longitudinal fluctuations (**B**) in all acute (n = 9) and (**C**) chronic (n = 5) GVHD episodes recorded in the study with at least one timepoint pre-GVHD onset. Namely, each GVHD episode was aligned on an imaginary “time “0”, splitting the “pre-GVHD” timepoints (boxed in blue and annotated on the X axis, with “negative” numbers referring to days prior to GVHD onset) from the onset of GVHD and timepoints thereafter (annotated with “positive” numbers on the X axis, referring to days post-onset). Individual (colored lines) and averaged (black line) gene expressions are displayed. Days pre- and post-GVHD onset on X axes are averaged across patient cohorts. (**D**) The same signatures were measured over the post-allo-HCT course in NO GVHD patients. The X axis for this group shows days after allo-HCT, averaged across patient cohorts. For a visual comparison of signatures over time between the GVHD and NO GVHD cohorts, each NO GVHD plot has a thick line on the bottom, showing the post-allo-HCT time frame when acute (in red) and chronic (in green) GVHD episodes in (**B**,**C**) occurred. The labeling “ns” in the pre-alloHCT samples indicates that these genes were not significantly different in patients pre transplantation compared with healthy donors.

**Table 1 cancers-17-00802-t001:** Patient characteristics of the analyzed cohort. AML: acute myeloid leukemia; MDS: myeloid dysplastic syndrome; ALL: acute lymphoblastic leukemia; CML: chronic myeloid leukemia; MF: myelofibrosis; HL: Hodgkin lymphoma.

Age	Years
Median	58
Range	22–72
Characteristics	N (%)
Sex	
Male	19 (61)
Female	12 (39)
Diagnosis	
AML	12 (39)
MDS	7 (23)
ALL	3 (12)
CML	2 (6)
MF	6 (19)
HL	1 (3)
Donor origin	
Related	10 (32)
Unrelated	21 (68)
Donor–recipient HLA match	
Matched	26 (84)
Mismatched	2 (6)
Haplo	3 (10)
Graft stem cell source	
Mobilized blood	28 (90)
Bone marrow	3 (10)

**Table 2 cancers-17-00802-t002:** List of patients included in the longitudinal analyses. Patients included in the longitudinal analyses with annotations regarding length of sampling, GVHD, infection, and relapse status during the analysis period. # In italics are shown the longitudinal samples with less than 10 timepoints that were excluded from CDP analysis. * Relapse events happened at day 315 for FHP1 and at day 445 for FHP4, in both cases after the last sample collected. RSV: respiratory syncytial virus; FLU: influenza virus; CMV: cytomegalovirus; TBC: tuberculosis; parainflu: parainfluenza virus; *Pseudom. A.*: *Pseudomonas aeruginosa*; *Clostr. Diff: Clostridium difficilis*; FUO: fever of unknown origin; *E. Coli*: *Escherichia coli*.

PT ID	Sample Series Length (Days After tx)	Total nr. of Samples Post-alloHCT	Group ID	Overall Max GVHD Grade	Infections	Relapse
BZP1	306	25	aGVHD	aGVHD I	RSV + FLU; CMV	no
BZP3	119	15	aGVHD	aGVHD III	TBC; *Pseudom. A.*	no
FHP26	290	19	aGVHD	aGVHD II	Suspected bacterial; Parainflu.	no
FHP38	188	17	aGVHD	aGVHD III	*Clostr. Diff*.	no
FHP42	95	10	aGVHD	aGVHD II	FUO; CMV	no
FHP34	215	16	aGVHD	aGVHD III	CMV; *E. Coli*	yes (d214)
FHP6	281	24	aGVHD and cGVHD	aGVHD II, cGVHD moderate	no	no
FHP1	244	16	aGVHD and cGVHD	aGVHD II, cGVHD mild	FUO	yes (d315) *
FHP11	324	14	aGVHD and cGVHD	aGVHD III, cGVHD mild	Parainflu.; *Clostr. Diff.*	yes (d157)
FHP4	430	33	aGVHD and cGVHD	aGVHD III, cGVHD moderate	CMV; Parainflu.	Yes (d445) *
*BZP5 #*	*70*	*8*	*aGVHD and cGVHD*	*aGVHD II*, *chronic mod/sev*	*EBV + bacterial infection*	*no*
BZP2	273	19	NO GVHD	none	no	no
*FHP32 #*	*78*	*8*	*NO GVHD*	*none*	*no*	*no*
FHP44	160	18	NO GVHD	none	no	no
*FHP46 #*	*73*	*3*	*NO GVHD*	*none*	*no*	*no*
median	215					
max	430					
min	70					

## Data Availability

All data used in the study are available in Appendix A. For further enquiries: sdeola@sidra.org.

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
