# Peer review of "Whole-Blood Longitudinal Molecular Profiling Maps the Road of Graft Versus Host Disease (GVHD)"

_cancers, 2025, doi:10.3390/cancers17050802_

Round 1
Reviewer 1 Report
Comments and Suggestions for Authors
The authors' work on the Molecular profiling of GVHD is commendable and could be a key reference for future studies. However, there are areas that need revision to enhance the manuscript's quality. I strongly urge the authors to address the concerns raised in this manuscript and consider the following comments and suggestions to improve their manuscript.
Blood sample collection is conducted weekly before the transplant and post-transplant every two weeks. But why, after the transplant, are there so many changes in biomarker expression? Why are the two-week samples preferred? There is also no significant change unless the patient is on medication or has made any dietary changes, such as testing for specific markers before the weekly samples pre-transplant. Samples are also analysed; is there a particular time for testing? There is no clear indication or explanation.
There are chances of more mRNA contamination in mobility blood transfer from the patient’s location to the testing location, which raises questions about how accurate the purity of RNA is.
Author Response
Comment 1: The authors' work on the Molecular profiling of GVHD is commendable and could be a key reference for future studies. However, there are areas that need revision to enhance the manuscript's quality. I strongly urge the authors to address the concerns raised in this manuscript and consider the following comments and suggestions to improve their manuscript.
Blood sample collection is conducted weekly before the transplant and post-transplant every two weeks.
Response 1: Actually, it is not exactly like that, and we realized that the description in the methods was confusing, leading to a misinterpretation, therefore we clarified (lines 109-110) that “Micro-quantities (50 mL) of whole blood were collected once prior to transplant conditioning, then weekly from engraftment time until day 100 post-transplantation and every 2 weeks thereafter until 2 years post-transplantation.
Comment 2: But why, after the transplant, are there so many changes in biomarker expression? Why are the two-week samples preferred?
Response 2: The main point of the project is to capture any “still unexplored” cause of GVHD, either preceding or coinciding with the onset of GVHD, to identify potential new diagnostic or predictive biomarkers. We think that this approach has a potential great value in a complication like GVHD that is still largely unpredictable as event per se, and as clinical outcome.
This goal was pursued in our study by linking the frequent blood analyses with the detailed clinical course of the patient. Early acute GVHD may happen right after engraftment, and acute GVHD happens usually in the first 3 months. We chose therefore for this period the 1-week schedule empirically, as a compromise between the highest possible frequence and feasibility for patients/analyses cost.
The same empirical schedule was applied after day 100 post – transplantation to cover late acute and chronic GVHD with the compromise of 2 weeks, allowing patients to follow a lighter schedule but still maintaining a high frequency of longitudinal analyses points.
We added an explanation at lines 111-113.
Comment 3: There is also no significant change unless the patient is on medication or has made any dietary changes, such as testing for specific markers before the weekly samples pre-transplant. Samples are also analysed; is there a particular time for testing? There is no clear indication or explanation.
Response 3: As mentioned before, the pre-transplant sample is only taken once.
Regarding the time of testing during the day, for both pre-transplant and post-transplant samples, we did not provide any indication to patients, other than specifying the date of the sampling on the vial, to be able to reconcile the RNA results precisely with the closest clinical annotations.
Comment 4: There are chances of more mRNA contamination in mobility blood transfer from the patient’s location to the testing location, which raises questions about how accurate the purity of RNA is.
Response 4: To prevent sample contamination during transfer, a double packaging system was implemented. The primary container was designed to be leak-proof and appropriately labeled to ensure sample integrity. It was then placed within a secondary container, which was securely sealed to provide an additional barrier against leaks and external contaminants. Throughout the transfer process, secure handling procedures were strictly followed, with transport personnel trained in biosafety protocols to mitigate any risks. Compliance with DOT/ICAO/IATA regulations ensured that the transport process met international safety standards, guaranteeing contamination-free and secure sample transfer.
To further verify the absence of contamination, qPCR reactions included non-template and blank negative controls, both of which showed no amplification. Any contamination on the packaging or sample vials would have resulted in a signal in the negative controls, but none was detected, confirming the integrity of the samples and the effectiveness of the handling procedures.
We added text at lines 142-144.

Reviewer 2 Report
Comments and Suggestions for Authors
Review for the manuscript ,,Whole-Blood Longitudinal Molecular Profiling Maps the Road of Graft Versus Host Disease (GVHD),,
In this study the authors want to develop a transcriptional fingerprint assay (TFA) in order to measure the abundance of 264 genes in patients after allogeneic hematopoietic cell transplantation.
The authors find that some genes such as that for “IFN”, “erythroid”, “protein synthesis”, “inflammation” “neutrophils/neutrophil activation” “cytokines/chemokines” transcriptional patterns allowed the identification of GVHD traits. One of the future perspective is to predict GVHD before their occurrence and to modulate therapeutic interventions.
The manuscript describes in detail the method used. The authors provide interesting data about the study. The results are presented in 2 tables and 5 figures. It would be recommended that tables have legends for abbreviations.
The results are clearly presented and discussed in detail in the Discussions chapter.
The study is useful to clinicians for an efficient management of this pathology.
Author Response
Comment 1: In this study the authors want to develop a transcriptional fingerprint assay (TFA) in order to measure the abundance of 264 genes in patients after allogeneic hematopoietic cell transplantation.
The authors find that some genes such as that for “IFN”, “erythroid”, “protein synthesis”, “inflammation” “neutrophils/neutrophil activation” “cytokines/chemokines” transcriptional patterns allowed the identification of GVHD traits. One of the future perspective is to predict GVHD before their occurrence and to modulate therapeutic interventions.
The manuscript describes in detail the method used. The authors provide interesting data about the study. The results are presented in 2 tables and 5 figures. It would be recommended that tables have legends for abbreviations.
The results are clearly presented and discussed in detail in the Discussions chapter.
The study is useful to clinicians for an efficient management of this pathology.
Response 1: We thank the reviewer for the kind appreciation of our work.
We added the abbreviations in the Tables legends.
Reviewer 3 Report
Comments and Suggestions for Authors
Congratulations to all the authors for this excellent and well-written paper on "Whole-blood longitudinal Molecular Profiling Maps the Road of GVHD". Given that peripheral blood gene expression is performed on blood samples which are generally larger in volumes drawn after venipuncture, the authors excellently describe here the implementation of Transcriptional Fingerprinting Assay (TFA) using multiplex microfluidics q-PCR capturing a wide range of systemic immune perturbations. The assay requires minimal volume of blood (50 microliters), which could be obtained by a microinvasive procedure such as finger stick, The authors in this study had a total of 31 samples pre-allo-HCT and 245 post-allo-HCT which were found to be suitable for analyses, collected until a median of 215 days after allo-HCT (70-430 days). The authors found that among patients who developed GVHD, three genes (GPM3, LRG1 and EPHX4) were significantly downregulated in the pre-transplant samples and two genes (IFI35 and PLK3) showed borderline (<1.5 fold) upregulation, although the change in gene expression did not reach the level of statistical significance. Twelve of these genes were highly upregulated (>4 fold) and their transcriptome was related to B Cells, T Lymphocytes, monocytes, IFN, TNF and other cytokines and neutrophil activation. The authors observed that these analyses highlighted clear signatures of DEGs associated with GVHD activity in different GVHD subtypes. The authors provided figures and tables. The main limitation of the study was the restricted number of subjects, particularly in the non-GVHD cohort and the lack of a validation cohort. The authors make a point that these findings increase the understanding of GVHD and reveal potentially targetable alterations, an approach which might be clinically implemented to clinically intercept GVHD before its occurrence and enable modulation of therapeutic intervention. The authors note that upon completion of this study, they aim to provide a comprehensive definition of the transcriptome signatures related to acute and chronic GVHD (cGVHD) and to validate the most important ones prospectively.
However, please note the following points:
1. There are few typographical mistakes. Please note on Page 2 Line 57: "While interesting transcriptome longitudinal data have been generates on animals". This may please be corrected - generated instead of generates.
2. Ocular involvement (oGVHD) is one of the most prevalent presentations of cGVHD which can manifest as dry eye disease, meibomian gland dysfunction, keratitis and conjunctivitis. The authors may wish to write in the manuscript whether or not they found any signatures which could detect oGVHD at earlier stages. Given that oGVHD is a subtype of chronic GVHD, early recognition of this could aide in their better management and prevention. The authors may wish to comment on this point. Thank you.
Thank you once again for writing this excellent manuscript on an important topic.
Author Response
Comment 1: Congratulations to all the authors for this excellent and well-written paper on "Whole-blood longitudinal Molecular Profiling Maps the Road of GVHD". Given that peripheral blood gene expression is performed on blood samples which are generally larger in volumes drawn after venipuncture, the authors excellently describe here the implementation of Transcriptional Fingerprinting Assay (TFA) using multiplex microfluidics q-PCR capturing a wide range of systemic immune perturbations. The assay requires minimal volume of blood (50 microliters), which could be obtained by a microinvasive procedure such as finger stick, The authors in this study had a total of 31 samples pre-allo-HCT and 245 post-allo-HCT which were found to be suitable for analyses, collected until a median of 215 days after allo-HCT (70-430 days). The authors found that among patients who developed GVHD, three genes (GPM3, LRG1 and EPHX4) were significantly downregulated in the pre-transplant samples and two genes (IFI35 and PLK3) showed borderline (<1.5 fold) upregulation, although the change in gene expression did not reach the level of statistical significance. Twelve of these genes were highly upregulated (>4 fold) and their transcriptome was related to B Cells, T Lymphocytes, monocytes, IFN, TNF and other cytokines and neutrophil activation. The authors observed that these analyses highlighted clear signatures of DEGs associated with GVHD activity in different GVHD subtypes. The authors provided figures and tables. The main limitation of the study was the restricted number of subjects, particularly in the non-GVHD cohort and the lack of a validation cohort. The authors make a point that these findings increase the understanding of GVHD and reveal potentially targetable alterations, an approach which might be clinically implemented to clinically intercept GVHD before its occurrence and enable modulation of therapeutic intervention. The authors note that upon completion of this study, they aim to provide a comprehensive definition of the transcriptome signatures related to acute and chronic GVHD (cGVHD) and to validate the most important ones prospectively.
However, please note the following points:
- There are few typographical mistakes. Please note on Page 2 Line 57: "While interesting transcriptome longitudinal data have been generates on animals". This may please be corrected - generated instead of generates.
Response 1: Corrected.
Comment 2: Ocular involvement (oGVHD) is one of the most prevalent presentations of cGVHD which can manifest as dry eye disease, meibomian gland dysfunction, keratitis and conjunctivitis. The authors may wish to write in the manuscript whether or not they found any signatures which could detect oGVHD at earlier stages. Given that oGVHD is a subtype of chronic GVHD, early recognition of this could aide in their better management and prevention. The authors may wish to comment on this point. Thank you.
Thank you once again for writing this excellent manuscript on an important topic.
Response 2: We thank the reviewer for the kind words and for appreciating our effort. This initial cohort of patients is not large enough to analyze subcategories of GVHD, but we fully agree on the importance of cGVHD early-stage prevention. Ocular cGVHD is indeed a common manifestation with often dramatic consequences on patients’ quality of life. While we hope to achieve more meaningful conclusions at the completion of the study, we added here in in Table S1 column “J” more specifications of the targeted cGVHD organs, including oGVHD. We hope the reviewer will find it useful, although we do not have the here the statistical power to perform an adequate analysis.
We also added comments in the text lines 556-561.